

# Reproducible, portable, and efficient ancient genome reconstruction with nf-core/eager

James A. Fellows Yates[1,2], Thiseas C. Lamnidis[1], Maxime Borry[1], Aida Andrades Valtueña[1], Zandra Fagernäs[1], Stephen Clayton[1], Maxime U. Garcia[3,4], Judith Neukamm[5,6] and Alexander Peltzer[1,7]

[1] Department of Archaeogenetics, Max Planck Institute for the Science of Human History, Jena, Germany
[2] Institut für Vor- und Frühgeschichtliche Archäologie und Provinzialrömische Archäologie, Ludwig-Maximilians-Universität München, München, Germany
[3] National Genomics Infrastructure, Science for Life Laboratory, Stockholm, Sweden
[4] Barntumörbanken, Department of Oncology-Pathology, Karolinska Institutet, Stockholm, Sweden
[5] Institute of Evolutionary Medicine, University of Zurich, Zurich, Switzerland
[6] Institute for Bioinformatics and Medical Informatics, Eberhard-Karls University Tübingen, Tübingen, Germany
[7] Quantitative Biology Center, Eberhard-Karls University Tübingen, Tübingen, Germany

## ABSTRACT

The broadening utilisation of ancient DNA to address archaeological, palaeontological, and biological questions is resulting in a rising diversity in the size of laboratories and scale of analyses being performed. In the context of this heterogeneous landscape, we present an advanced, and entirely redesigned and extended version of the EAGER pipeline for the analysis of ancient genomic data. This Nextflow pipeline aims to address three main themes: accessibility and adaptability to different computing configurations, reproducibility to ensure robust analytical standards, and updating the pipeline to the latest routine ancient genomic practices. The new version of EAGER has been developed within the nf-core initiative to ensure high-quality software development and maintenance support; contributing to a long-term life-cycle for the pipeline. nf-core/eager will assist in ensuring that a wider range of ancient DNA analyses can be applied by a diverse range of research groups and fields.

## INTRODUCTION

Ancient DNA (aDNA) has become a widely accepted source of biological data, helping to provide new perspectives for a range of fields including archaeology, cultural heritage, evolutionary biology, ecology, and palaeontology. The utilisation of short-read high-throughput sequencing has allowed the recovery of whole genomes and genome-wide data from a wide variety of sources, including (but not limited to), the skeletal remains of animals (*Palkopoulou et al., 2015*; *Orlando et al., 2013*; *Frantz et al., 2019*; *Star et al., 2017*), modern and archaic humans (*Damgaard et al., 2018*; *Green et al, 2010*; *Meyer et al, 2012*; *Slon et al, 2018*)-rv, bacteria (*Bos et al., 2014*; *Namouchi et al., 2018*;

Corresponding authors
James A. Fellows Yates,
fellows@shh.mpg.de
Alexander Peltzer,
peltzer@shh.mpg.de

*Schuenemann et al., 2018*), viruses (*Mühlemann et al., 2018*; *Krause-Kyora et al., 2018*), plants (*Wales et al., 2019*; *Gutaker et al., 2019*), palaeofaeces (*Tett et al., 2019*; *Borry et al., 2020*), dental calculus (*Warinner et al., 2014*; *Weyrich et al., 2017*), sediments (*Willerslev et al., 2014*; *Slon et al., 2017*), medical slides (*Van Dorp et al., 2019*), parchment (*Teasdale et al., 2015*), and recently, ancient 'chewing gum' (*Jensen et al., 2019*; *Kashuba et al., 2019*). Improvement in laboratory protocols to increase yields of otherwise trace amounts of DNA has at the same time led to studies that can total hundreds of ancient individuals (*Olalde et al., 2018*; *Mathieson et al., 2018*), spanning single (*Bos et al., 2011*) to thousands of organisms (*Warinner et al., 2014*). These differences of disciplines have led to a heterogeneous landscape in terms of the types of analyses undertaken, and their computational resource requirements (*Tastan Bishop et al., 2015*; *Bah et al, 2018*). Taking into consideration the unequal distribution of resources (and infrastructure such as internet connection), easy-to-deploy, streamlined and efficient pipelines can help increase accessibility to high-quality analyses.

The degraded nature of aDNA poses an extra layer of complexity to standard modern genomic analysis. Through a variety of processes (*Lindahl, 1993*) DNA molecules fragment over time, resulting in ultra-short molecules (*Meyer et al., 2016*). These sequences have low nucleotide complexity making it difficult to identify with precision which part of the genome a read (a sequenced DNA molecule) is derived from. Fragmentation without a 'clean break' leads to uneven ends, consisting of single-stranded 'overhangs' at ends of molecules that are susceptible to chemical processes such as deamination of nucleotides. These damaged nucleotides then lead to misincorporation of complementary bases during library construction for high-throughput DNA sequencing (*Briggs et al., 2007*). On top of this, taphonomic processes such as heat, moisture, and microbial- and burial-environment processes lead to varying rates of degradation (*Kistler et al., 2017*; *Warinner et al., 2017*). The original DNA content of a sample is therefore increasingly lost over time and supplanted by younger 'environmental' DNA. Later handling by archaeologists, museum curators, and other researchers can also contribute 'modern' contamination. While these characteristics can help provide evidence towards the 'authenticity' of true aDNA sequences (e.g., the aDNA cytosine to thymine or C to T 'damage' deamination profiles as by *Ginolhac et al., 2011*), they also pose specific challenges for genome reconstruction, such as unspecific DNA alignment and/or low coverage and miscoding lesions that can result in low-confidence genotyping. These factors often lead to prohibitive sequencing costs when retrieving enough data for modern high-throughput short-read sequencing data pipelines (such as more than 1 billion reads for a 1X depth coverage *Yersinia pestis* genome, as in *Rasmussen et al., 2015*), and thus aDNA-tailored methods and techniques are required to overcome these challenges.

Two previously published and commonly used pipelines in the field are PALE-OMIX (*Schubert et al., 2014*) and EAGER (*Peltzer et al., 2016*). These two pipelines take a similar approach to link together standard tools used for Illumina high-throughput short-read data processing (sequencing quality control, sequencing adapter removal and/or paired-end read merging, mapping of reads to a reference genome, genotyping, etc.). However, they have a specific focus on tools that are designed for, or well-suited

for aDNA (such as the bwa aln algorithm for ultra-short molecules (*Li & Durbin, 2009*) and mapDamage (*Jónsson et al., 2013*) for evaluation of aDNA characteristics). Yet, neither of these genome reconstruction pipelines have had major updates to bring them in-line with current routine bioinformatic practices (such as continuous integration tests and software containers) and aDNA analyses. In particular, *Meta*genomic screening of off-target genomic reads for pathogens or microbiomes (*Warinner et al., 2014*; *Weyrich et al., 2017*) has become common in palaeo- and archaeogenetics, given its role in revealing widespread infectious disease and possible epidemics that have sometimes been previously undetected in the archaeological record (*Mühlemann et al., 2018*; *Krause-Kyora et al., 2018*; *Rasmussen et al., 2015*; *Andrades Valtueña et al., 2017*). Without easy access to the latest field-established analytical routines, ancient genomic studies risk being published without the necessary quality control checks that ensure aDNA authenticity, as well as limiting the full range of possibilities from their data. Given that material from samples is limited, there are both ethical as well as economical interests to maximise analytical yield (*Green & Speller, 2017*).

To address these shortcomings, we have completely re-implemented the latest version of the EAGER pipeline in Nextflow (*Di Tommaso et al., 2017*) (a domain-specific-language or 'DSL', specifically designed for the construction of omics analysis pipelines), and introduced new features and more flexible pipeline configuration. In addition, the renamed pipeline—nf-core/eager—has been developed in the context of the nf-core community framework (*Ewels et al., 2020*), which enforces strict guidelines for best-practices in software development.

## MATERIALS AND METHODS

### Updated Workflow

The new pipeline follows a similar structural foundation to the original version of EAGER (Fig. 1) and partially to PALEOMIX. Given Illumina short-read FASTQ and/or BAM files and a reference FASTA file, the core functionality of nf-core/eager can be split into five main stages:

1. Pre-processing:

   - Sequencing quality control: FastQC (*Andrews, 2010*)
   - Sequencing artefact clean-up (merging, adapter clipping): AdapterRemoval2 (*Schubert, Lindgreen & Orlando, 2016*), fastp (*Chen et al., 2018*)
   - Pre-processing statistics generation: FastQC

2. Mapping and post-processing:

   - Alignment against reference genome: BWA aln and mem (*Li & Durbin, 2009*; *Li, 2013*), CircularMapper (*Peltzer et al., 2016*), Bowtie2 (*Langmead & Salzberg, 2012*)
   - Mapping quality filtering: SAMtools (*Li et al., 2009*)
   - PCR duplicate removal: Picard MarkDuplicates (http://broadinstitute.github.io/picard/), DeDup (*Peltzer et al., 2016*)
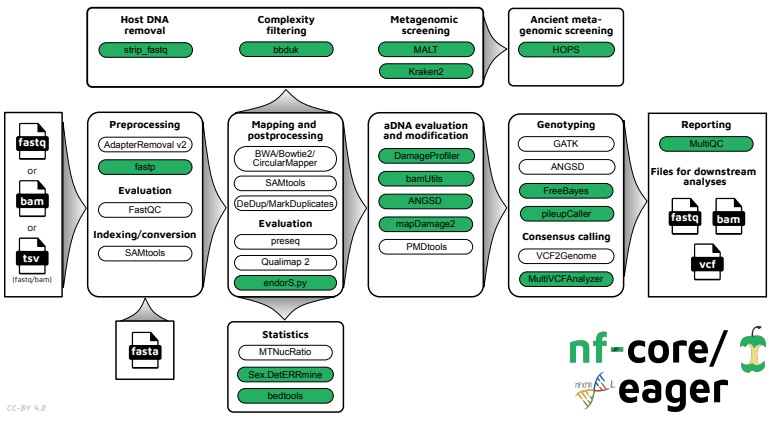

**Figure 1** **Simplified schematic of the nf-core/eager workflow pipeline.** Green filled bubbles indicate new functionality added over the original EAGER pipeline.

- Mapping statistics generation: SAMtools, PreSeq (*Daley & Smith, 2013*), Qualimap2 (*Okonechnikov, Conesa & García-Alcalde, 2016*), bedtools (*Quinlan & Hall, 2010*), Sex.DetERRmine (*Lamnidis et al., 2018*)

3. aDNA evaluation and modification:

   - Damage profiling: DamageProfiler (*Neukamm, Peltzer & Nieselt, 2020*)
   - aDNA reads selection: PMDtools (*Skoglund et al., 2014*)
   - Damage removal/Base trimming: mapDamage2 (*Jónsson et al., 2013*), Bamutils (*Jun et al., 2015*)
   - Human nuclear contamination estimation: ANGSD (*Korneliussen, Albrechtsen & Nielsen, 2014*)

4. Variant calling and consensus sequence generation: GATK UnifiedGenotyper and HaplotypeCaller (*McKenna et al., 2010*), sequenceTools pileupCaller (https://github.com/stschiff/sequenceTools), VCF2Genome (*Peltzer et al., 2016*), MultiVCFAnalyzer (*Bos et al., 2014*)

5. Report generation: MultiQC (*Ewels et al., 2016*)

In nf-core/eager, all tools originally used in EAGER have been updated to their latest versions, as available on Bioconda (*Grüning et al., 2018*) and conda-forge (https://github.com/conda-forge), to ensure widespread accessibility and stability of utilised tools. The mapDamage2 (for damage profile generation) (*Jónsson et al., 2013*) and Schmutzi (for mitochondrial contamination estimation) (*Renaud et al., 2015*) methods have not been carried over to nf-core/eager, the first because a faster successor method is now available (DamageProfiler, *Neukamm, Peltzer & Nieselt, 2020*), and the latter because a stable release of the method could not be migrated to Bioconda at time of writing. We anticipate that there will be an updated version of Schmutzi in the near future that will allow us to integrate the method again into nf-core/eager. As an alternative, estimation of human *nuclear* contamination is now offered through ANGSD. mapDamage2 is however retained

to offer probabilistic *in silico* damage removal from BAM files. Support for the Bowtie2 aligner has been updated to have default settings optimised for aDNA (*Poullet & Orlando, 2020*).

New tools to the basic workflow include fastp for the removal of 'poly-G' sequencing artefacts that are common in 2-colour Illumina sequencing machines (such as the increasingly popular NextSeq and NovaSeq platforms). For variant calling, we have now included FreeBayes (*Garrison & Marth, 2012*) as an alternative to the human-focused GATK tools, and have also added pileupCaller for generation of genotyping formats commonly utilised in ancient human population analysis. We have also maintained the possibility of using the now officially unsupported GATK UnifiedGenotyper, as the supported replacement, GATK HaplotypeCaller, performs *de novo* assembly around possible variants; something that may not be suitable for low-coverage aDNA data.

Additional functionality tailored for ancient bacterial genomics includes integration of a SNP alignment generation tool, MultiVCFAnalyzer, which includes the ability to make an assessment of levels of cross-mapping from different related taxa to a reference genome - a common challenge in ancient bacterial genome reconstruction (as discussed in *Warinner et al., 2017*). The output SNP consensus alignment FASTA file can then be used for downstream analyses such as phylogenetic tree construction. Simple coverage statistics of particular annotations (e.g., genes) of an input reference is offered by bedtools, which can be used in cases such as for providing initial indications of functional differences between ancient bacterial strains (as in *Andrades Valtueña et al., 2017*). For analysis of human genomes, nf-core/eager can also give estimates of the relative coverage on the X and Y chromosomes with Sex.DetERRmine, which can be used to infer the biological sex of a given human individual. A dedicated 'endogenous DNA' calculator (endorS.py) is also included, to provide a percentage estimate of the sequenced reads matching the reference ('on-target') from the total number of reads sequenced per library.

Given the large amount of sequencing often required to yield sufficient genome coverage from aDNA data, palaeogenomicists tend to use multiple (differently treated) libraries, and/or merge data from multiple sequencing runs of each library or even samples. The original EAGER pipeline could only run a single library at a time, and in these contexts required significant manual user input in merging different FASTQ or BAM files of related libraries. A major upgrade in nf-core/eager is that the new pipeline supports automated processing of complex sequencing strategies for many samples, similar to PALEOMIX. This is facilitated by the optional use of a simple table (in TSV format, a format more commonly used in wet-lab stages of data generation, compared to PALEOMIX's YAML format) that includes file paths and additional metadata such as sample name, library name, sequencing lane, colour chemistry, and UDG treatment. This allows automated and simultaneous processing and appropriate merging and treatment of heterogeneous data from multiple sequencing runs and/or library types (Fig. 2).

The original EAGER and PALEOMIX pipelines required users to look through many independent output directories and files to make full assessment of their sequencing data. This has now been replaced in nf-core/eager with a much more extensive MultiQC report. This tool aggregates the log files of every supported tool into a single interactive report, and
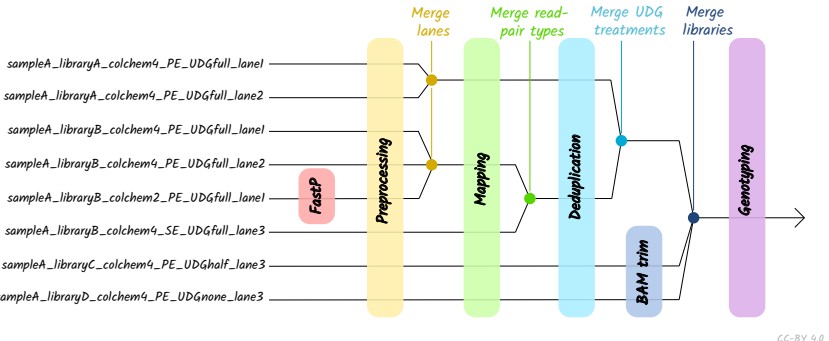

**Figure 2** **Diagram of different processing and library-merging points based on the nature of different libraries.** Merge points represent merging of related BAM files as defined by metadata fields in an input TSV file. 'colchem' refers to the colour chemistry system of Illumina sequencers (2 for e.g., NextSeq or 4 for HiSeq machines). 'PE/SE' refers to paired-end and single-end sequencing chemistries. 'UDG' refers to uracil DNA glycosylase treatment, a laboratory procedure to completely or partially remove C to T mis-coding lesions. Lane refers to sequencing lane.

assists users in making a fuller assessment of their sequencing and analysis runs. We have developed a corresponding MultiQC module for every tool used by nf-core/eager, where possible, to enable comprehensive evaluation of all stages of the pipeline.

We have further extended the functionality of the original EAGER pipeline by adding ancient metagenomic analysis (Fig. 3); allowing reconstruction of the wider taxonomic content of a sample. We have added the possibility to screen all off-target reads (not mapped to the reference genome) with two metagenomic profilers: MALT (*Herbig et al., 2016*; *Vågene et al., 2018*) and Kraken2 (*Wood, Lu & Langmead, 2019*), in parallel to the mapping to a given reference genome (typically of the host individual, assuming the sample is a host organism). Pre-profiling removal of low-sequence-complexity reads that can slow down profiling and result in false-positive taxonomic identifications is offered through BBduk (Brian Bushnell: http://sourceforge.net/projects/bbmap/). Post-profiling characterisation of properties of authentic aDNA from metagenomic MALT alignments is carried out with MaltExtract of the HOPS pipeline (*Hübler et al., 2019*). This functionality can be used either for microbiome screening or putative pathogen detection. Ancient metagenomic studies sometimes include comparative samples from living individuals (*Velsko et al., 2019*). To support open data, whilst respecting personal data privacy, nf-core/eager includes a 'FASTQ host removal' script that creates raw FASTQ files, but with all reads successfully mapped to the reference genome removed. This allows for safe upload of metagenomic non-host sequencing data to public repositories after removal of identifiable (human) data, for example for microbiome studies.

An overview of the entire pipeline is shown in Fig. 1, and a tabular comparison of functionality between EAGER, PALEOMIX and nf-core/eager is in Table 1.

## Usage

nf-core/eager can be run on POSIX-family operating systems (e.g., Linux and macOS) and has at minimum three dependencies, Java (>=8), Nextflow (*Di Tommaso et al., 2017*),

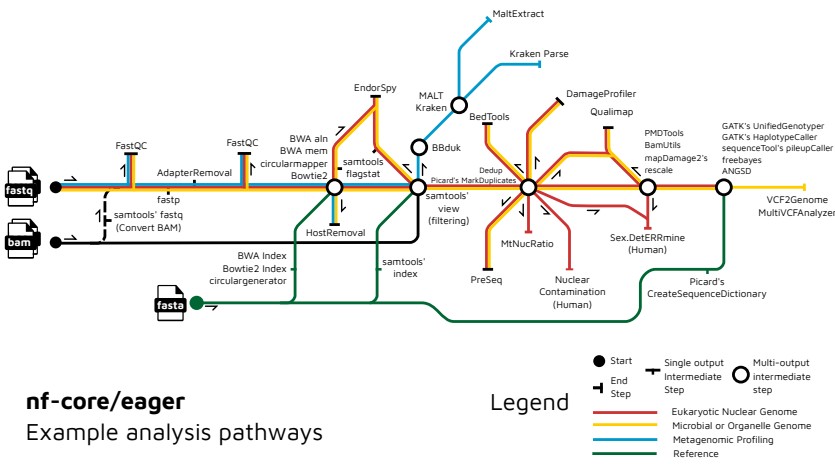

**Figure 3   Example routes of analyses offered by nf-core/eager.** nf-core/eager includes functionality that is applicable to different common aDNA research contexts, such as for population genetics, microbial genome reconstruction, and metagenomic screening of ancient microbiomes.

and a software container or environment system, e.g., Conda (https://conda.io), Docker (https://www.docker.com) or Singularity (https://sylabs.io), and can be run on both local machines as well as high performance computing (HPC) clusters. Once installed, Nextflow handles all subsequent pipeline-related software dependencies.

The pipeline is primarily a command-line interface (CLI), and therefore users execute the pipeline with a single command, and can specify additional options via parameters and flags. Alternatively, the nf-core initiative offers a graphical user interface (GUI) for this less familiar with CLIs (via https://nf-co.re/launch). Routinely used parameters or institutional configurations can be specified using Nextflow's in-built profile system, an example of which is provided in the basic test data, which can also be used for training and evaluation by new users. This allows specification of many pipeline parameters but with a single option. Specific versions of the pipeline or even specific git hash commits can be supplied as a parameter to ensure analytical reproducibility by other researchers.

The pipeline accepts as input raw short-read FASTQ files, BAM files, or a sample sheet in TSV format that contains extra metadata, and a reference genome in FASTA format. Usage of a TSV file allows processing of more complex sequencing strategies and heterogeneous data types - such as per-run processing of libraries sequenced across different models of sequencing machines, or customised per-library clipping depending on laboratory UDG-treatment (*Rohland et al., 2015*, Fig. 2). Additional files can be supplied for efficiency, such as pre-built reference indices, or annotation files, depending on the modules that wish to be run. Monitoring of pipeline progress is primarily performed via on-console logging.

By default, nf-core/eager performs sequencing QC, adapter clipping (and assuming paired-end data, merging), aligning of the processed reads to the supplied reference genome, PCR duplicate removal and finally aDNA damage and mapping-quality evaluation. If running on a HPC cluster utilising a scheduling system, Nextflow will generate and carry

**Table 1** Comparison of pipeline functionality of common ancient DNA processing pipelines.

| Category | Functionality | EAGER | PALEOMIX | nf-core/eager |
|---|---|---|---|---|
| Infrastructure | Software environments | Yes | No | Yes |
| | HPC scheduler integration | No | No | Yes |
| | Cloud computing integration | No | No | Yes |
| | Per-process resource optimisation | No | Partial | Yes |
| | Pipeline-step parallelisation | No | Yes | Yes |
| | Command line set up | No | Yes | Yes |
| | GUI set up | Yes | No | Yes |
| Preprocessing | Sequencing lane merging | Yes | Yes | Yes |
| | Sequencing quality control | Yes | No | Yes |
| | Sequencing artefact removal | No | No | Yes |
| | Adapter clipping/read merging | Yes | Yes | Yes |
| | Post-processing sequencing QC | No | No | Yes |
| Alignment | Reference mapping | Yes | Yes | Yes |
| | Reference mapping statistics | Yes | Yes | Yes |
| | Multi-reference mapping | No | Yes | No |
| Postprocessing | Mapped reads filtering | Yes | Yes | Yes |
| | Metagenomic complexity filtering | No | No | Yes |
| | Metagenomic profiling | No | No | Yes |
| | Metagenomic authentication | No | No | Yes |
| | Library complexity estimation | Yes | No | Yes |
| | Duplicate removal | Yes | No | Yes |
| | BAM merging | No | Yes | Yes |
| Authentication | Damage read filtering | Yes | No | Yes |
| | Human contamination estimation | Yes | No | Yes |
| | Human biological sex determination | No | No | Yes |
| | Genome coverage estimation | Yes | Yes | Yes |
| | Damage calculation | Yes | Yes | Yes |
| | Damage rescaling | No | Yes | Yes |
| Downstream | SNP calling/genotyping | Yes | Partial | Yes |
| | Consensus sequence generation | Yes | Partial | Yes |
| | Regions of interest statistics | Partial | Yes | Yes |

out efficient submission strategies of jobs for the user. The pipeline produces a multitude of output files in various file formats, with a more detailed listing available in the user documentation. These include metrics, statistical analysis data, and standardised output files (BAM, VCF) for close inspection and further downstream analysis, as well as a MultiQC report. If an emailing daemon is set up on the server, the latter can be emailed to users automatically.

## Benchmarking
### *Functionality demonstration*

To demonstrate the simultaneous genomic analysis of human DNA and metagenomic screening for putative pathogens, as well as improved results reporting, we re-analysed data from *Barquera et al. (2020)* who performed a multi-discipline study of three 16th century individuals excavated from a mass burial site in Mexico City. The authors reported genetic results showing sufficient on-target human DNA (>1%) with typical aDNA damage (>20% C to T reference mismatches in the first base of the 5′ ends of reads) for downstream population-genetic analysis and Y-chromosome coverage indicative that the three individuals were genetically male. In addition, one individual (Lab ID: SJN003) contained DNA suggesting a possible infection by *Treponema pallidum*, a species with a variety of strains that can cause diseases such as syphilis, bejel and yaws, and a second individual (Lab ID: SJN001) displayed reads similar to the Hepatitis B virus. Both results were confirmed by the authors via in-solution enrichment approaches.

Full step-by-step instructions on the setup of the human and pathogen screening demonstration (including input TSV file and final command) can be seen in Data S1. In brief, we replicated the results of *Barquera et al. (2020)* using nf-core/eager v2.2.0 (commit: e7471a7 and Nextflow version: 20.04.1) by simultaneously aligning publicly available shotgun-sequencing reads against the human reference genome (hs37d5) using bwa aln and the off-target reads against the NCBI Nucleotide (nt) database (October 2017 - uploaded here to Zenodo under DOI: 10.5281/zenodo.4382153) with MALT. Alignment parameters were as close to as reported in the original publication, otherwise kept as default. The modified parameter values for pathogen detection were used, rather than nf-core/eager defaults, as these parameters can be highly target-species dependent and must be modified on a per-context basis. Additional modules turned on were mitochondrial-to-nuclear ratio calculation, nuclear contamination estimation, and biological sex determination.

To include the HOPS results from metagenomic screening in the report, we also re-ran MultiQC with the upcoming version v1.10 (to be integrated into nf-core/eager on release), which has an integrated HOPS module. After installing the development version of MultiQC (commit: 7584e64), as described in the MultiQC documentation (https://multiqc.info/), we re-ran the MultiQC command used with the pipeline.

### *Run-time comparison*

We also compared pipeline run-times of two functionally equivalent and previously published pipelines to show that the new implementation of nf-core/eager is equivalent or more efficient than EAGER or PALEOMIX. We ran each pipeline on a subset of Viking-age genomic data of cod (*Gadus morhua*) from *Star et al. (2017)*. This data was originally run using PALEOMIX, and was re-run here as described, but with the latest version of PALEOMIX (v1.2.14), and with equivalent settings for the other two pipelines as close as possible to the original paper (EAGER with v1.92.33, and nf-core/EAGER with v2.2.0, commit 830c22d).

The respective benchmarking environment and exact pipeline run settings can be seen in the Data S1. Two samples each with three Illumina paired-end sequencing runs

were analysed, with adapter clipping and merging (AdapterRemoval), mapping (BWA aln), duplicate removal (Picard's MarkDuplicates) and damage profiling (PALEOMIX: mapDamage2, EAGER and nf-core/EAGER: DamageProfiler) steps being performed. Run-times comparisons were performed on a 32 CPU (AMD Opteron 23xx) and 256 GB memory Red Hat QEMU Virtual Machine running the Ubuntu 18.04 operating system (Linux Kernel 4.15.0-112). Resource parameters of each tool were only modified to specify the maximum available on the server and otherwise left as default. We ran the commands for each tool sequentially, but repeated these batches of commands 10 times - to account for variability in the cloud service's IO connection. Run times were measured using the GNU time tool (v1.7).

## RESULTS

### Functionality demonstration

We were able to successfully replicate the human and pathogen screening results in a single run of nf-core/eager. Mapping to the human reference genome (hs37d5) with BWA aln and binning of off-target reads with MALT to the NCBI Nucleotide database (2017-10-26), yielded the same results of all individuals having a biological sex of male, as well as the same frequency of C to T miscoding lesions and short mean fragment lengths (both characteristic of true aDNA). Metagenomic hits to both pathogens from the corresponding individuals that yielded complete genomes in the original publication were also detected. Both results and other processing statistics were identified via a single interactive MultiQC report, excerpts of which can be seen in Fig. 4. The full interactive report can be seen in the Data S1.

### Run-time comparison

A summary of run-times of the benchmarking tests can be seen in Table 2. nf-core/eager showed fastest run-times across all three time metrics when running on default parameters. This highlights the improved efficiency of nf-core/eager's asynchronous processing system and per-process resource customisation (here represented by nf-core/eager defaults designed for typical HPC cluster setup).

As a more realistic demonstration of modern computing multi-threading setup, we also re-ran PALEOMIX with the flag –max-bwa-threads set to 4 (listed in Table 2 as 'optimised'), which is equivalent to a single BWA aln process of nf-core/eager. This resulted in a much faster run-time than that of default nf-core/eager, due to the approach of PALEOMIX of mapping each lane of a library separately, whereas nf-core/eager will map all lanes of a single library merged together. Therefore, given that each library was split across three lanes, increasing the threads of BWA aln to 4 resulted in 12 per library, whereas nf-core/eager only gave 4 (by default) for a single BWA aln process of one library. While the PALEOMIX approach is valid, we opted to retain the per-library mapping as it is often the longest running step of high-throughput sequencing genome-mapping pipelines, and it prevents flooding of HPC scheduling systems with many long-running jobs. Secondly, if users regularly use multi-lane data, due to nf-core/eager's fine-granularity control, they can simply modify nf-core/eager's BWA aln process resources via config files to account

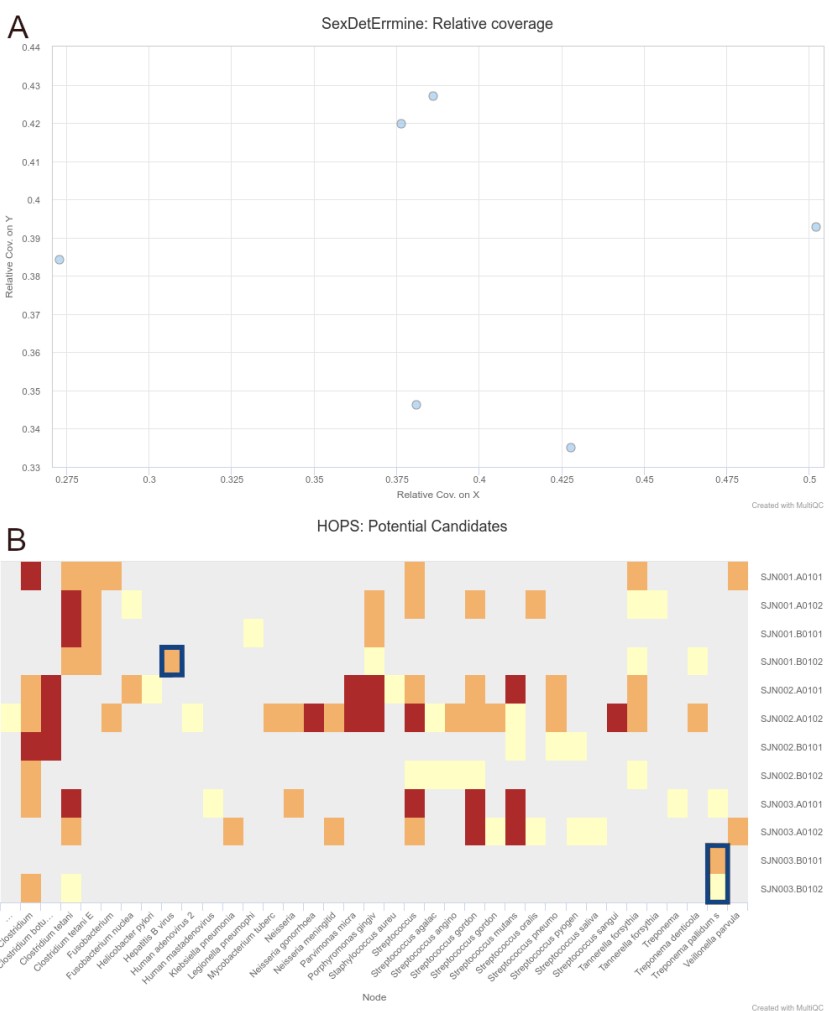

**Figure 4** Sections of a MultiQC report (v1.10dev) with the outcome of simultaneous human DNA and microbial pathogen screening with nf-core/eager, including (A) Sex.DetERRmine output of biological sex assignment with coverages on X and Y being half of that of autosomes, indicative of male individuals, and (B) HOPS output with positive detection of both *Treponema pallidum* and Hepatitis B virus reads (indicated with blue boxes). Other taxa in HOPS output represent typical environmental contamination and oral commensal microbiota found in archaeological teeth. Data was Illumina shotgun sequencing data from *Barquera et al. (2020)*, and replicated results here were originally verified in the publication via enrichment methods. The full interactive reports for both MultiQC v1.9 and v1.10 can be seen in Data S1.

for this. When we optimised parameters that were used for BWA aln's multi-threading, and the number of multiple lanes to the same number of BWA aln threads as the optimised PALEOMIX run, nf-core/eager again displayed faster run-times (Table 2).

All metrics including mapped reads, percentage on-target, mean depth coverage and mean read lengths across all pipeline and replicates were extremely similar (Table 3).

**Table 2** **Comparison of run-times in minutes between three ancient DNA pipelines.** PALEOMIX and nf-core/eager have additional runs with 'optimised' parameters with fairer computational resources matching modern multi-threading strategies. Values represent mean and standard deviation of run-times in minutes, calculated from the output of the GNU time tool. Real: real time, System: cumulative CPU system-task times, User: cumulative CPU time of all tasks.

| Pipeline | Version | Environment | real | sys | user |
|---|---|---|---|---|---|
| nf-core-eager (optimised) | 2.2.0dev | singularity | 105.6 ± 4.6 | 13.6 ± 0.7 | 1593 ± 79.7 |
| PALEOMIX (optimised) | 1.2.14 | conda | 130.6 ± 8.7 | 12 ± 0.7 | 1820.2 ± 36.9 |
| nf-core-eager | 2.2.0dev | singularity | 209.2 ± 4.4 | 11 ± 0.9 | 1407.7 ± 30.2 |
| EAGER | 1.92.37 | singularity | 224.2 ± 4.9 | 22.9 ± 0.3 | 1736.3 ± 70.2 |
| PALEOMIX | 1.2.14 | conda | 314.6 ± 2.9 | 10.7 ± 1 | 1506.7 ± 14 |

**Table 3** **Comparison of output statistics between three ancient DNA pipelines.** Comparison of common result values of key high-throughput short-read data processing and mapping steps across the three pipelines, as reported by the equivalent value of the report from each tool. 'QF' stands for mapping-quality filtered reads. All values represent mean and standard deviation across 10 replicates of each pipeline.

| Sample | Category | EAGER | nf-core/eager | PALEOMIX |
|---|---|---|---|---|
| COD076 | Processed Reads | 71,388,991 ± 0 | 71,388,991 ± 0 | 72,100,142 ± 0 |
| COD092 | Processed Reads | 69,615,709 ± 0 | 6,9615,709 ± 0 | 70,249,181 ± 0 |
| COD076 | Mapped QF Reads | 16,786,467.7 ± 106.5 | 16,786,491.1 ± 89.9 | 16,686,607.2 ± 91.3 |
| COD092 | Mapped QF Reads | 16,283,216.3 ± 71.3 | 16,283,194.7 ± 37.4 | 16,207,986.2 ± 44.4 |
| COD076 | Percent QF On-target | 23.5 ± 0 | 23.5 ± 0 | 23.1 ± 0 |
| COD092 | Percent QF On-target | 23.4 ± 0 | 23.4 ± 0 | 23.1 ± 0 |
| COD076 | Deduplicated Reads | 12,107,264.4 ± 87.8 | 12,107,293.7 ± 69.7 | 12,193,415.8 ± 86.7 |
| COD092 | Deduplicated Reads | 13,669,323.7 ± 87.6 | 13,669,328 ± 32.4 | 13,795,703.3 ± 47.9 |
| COD076 | Mean Depth Coverage | 0.9 ± 0 | 0.9 ± 0 | 0.9 ± 0 |
| COD092 | Mean Depth Coverage | 1 ± 0 | 1 ± 0 | 1 ± 0 |
| COD076 | Mean Read Length | 49.4 ± 0 | 49.4 ± 0 | 49.4 ± 0 |
| COD092 | Mean Read Length | 48.8 ± 0 | 48.8 ± 0 | 48.7 ± 0 |

## DISCUSSION

The re-implementation of EAGER into Nextflow offers a range of benefits over the original custom pipeline framework.

The new framework provides immediate integration of nf-core/eager into various job schedulers in POSIX HPC environments, cloud computing resources, as well as local workstations. This portability allows users to set up nf-core/eager regardless of the type of computing infrastructure or cluster size (if applicable), with minimal effort or configuration. This facilitates reproducibility and therefore maintenance of standards within the field. Portability is further assisted by the in-built compatibility with software environments and containers such as Conda, Docker and Singularity. These are isolated software 'sandbox' environments that include all software (with exact versions) required by the pipeline, in a form that is installable and runnable by users regardless of the setup of their local software environment. Another major change with nf-core/eager is that the primary user interaction mode of a pipeline run setup is now with a CLI, replacing the GUI of the original EAGER pipeline. This is more portable and compatible with most

HPC clusters (that may not offer display of a window system), and is in line with the vast majority of bioinformatics tools. We therefore believe this will not be a hindrance to new researchers from outside computational biology. However, a GUI-based pipeline set up is still available via the nf-core website's Launch page (https://nf-co.re/launch), which provides a common GUI format across multiple pipelines, as well as additional robustness checks of input parameters for those less familiar with CLIs. Typically the output of the launch functionality is a JSON file that can be used with a nf-core/tools launch command as a single parameter (similar to the original EAGER), however integration with Nextflow's companion monitoring tool tower.nf (https://tower.nf) also allows direct submission of pipelines without any command line usage.

Reproducibility is made easier through the use of 'profiles' that can define configuration parameters. These profiles can be managed at different hierarchical levels. *HPC cluster-level profiles* can specify parameters for the computing environment (job schedulers, cache locations for containers, maximum memory and CPU resources etc.), which can be centrally managed to ensure all users of a group use the same settings. *Pipeline-level profiles*, specifying parameters for nf-core/eager itself, allow fast access to routinely-run pipeline parameters via a single flag in the nf-core/eager run command, without having to configure each new run from scratch. Compared to the original EAGER, which utilised per-FASTQ XML files with hardcoded filepaths for a specific user's server, nf-core/eager allows researchers to publish the specific profile used in their runs alongside their publications, which can also be used by other groups to generate the same results. Usage of profiles can also reduce mistakes caused by insufficient 'prose' based reporting of program settings that can be regularly found in the literature. The default nf-core/eager profile uses parameters evaluated in different aDNA-specific contexts (e.g., in *Poullet & Orlando, 2020*), and will be updated in each new release as new studies are published.

nf-core/eager provides improved efficiency over the original EAGER pipeline by replacing sample-by-sample sequential processing with Nextflow's asynchronous job parallelisation, whereby multiple pipeline steps and samples are run in parallel (in addition to natively parallelised pipeline steps). This is similar to the approach taken by PALEOMIX, however nf-core/eager expands this by utilising Nextflow's ability to customise the resource parameters for every job in the pipeline; reducing unnecessary resource allocation that can occur with unfamiliar users to each step of a high-throughput short-read data processing pipeline. This is particularly pertinent given the increasing use of centralised HPC clusters or cloud computing that often use per-hour cost calculations.

Alongside the interactive MultiQC report, we have written extensive documentation on all parts of running and interpreting the output of the pipeline. Given that a large fraction of aDNA researchers come from fields outside computational biology, and thus may have limited computational training, we have written documentation and tutorials (https://nf-co.re/eager/) that also give guidance on how to run the pipeline and interpret each section of the report in the context of high-throughput sequencing data, but with a special focus on aDNA. This includes best practice or expected output schematic images that are published under CC-BY licenses to allow for use in other training material (an example can be seen in Fig. 5). We hope this open-access resource will make the study of

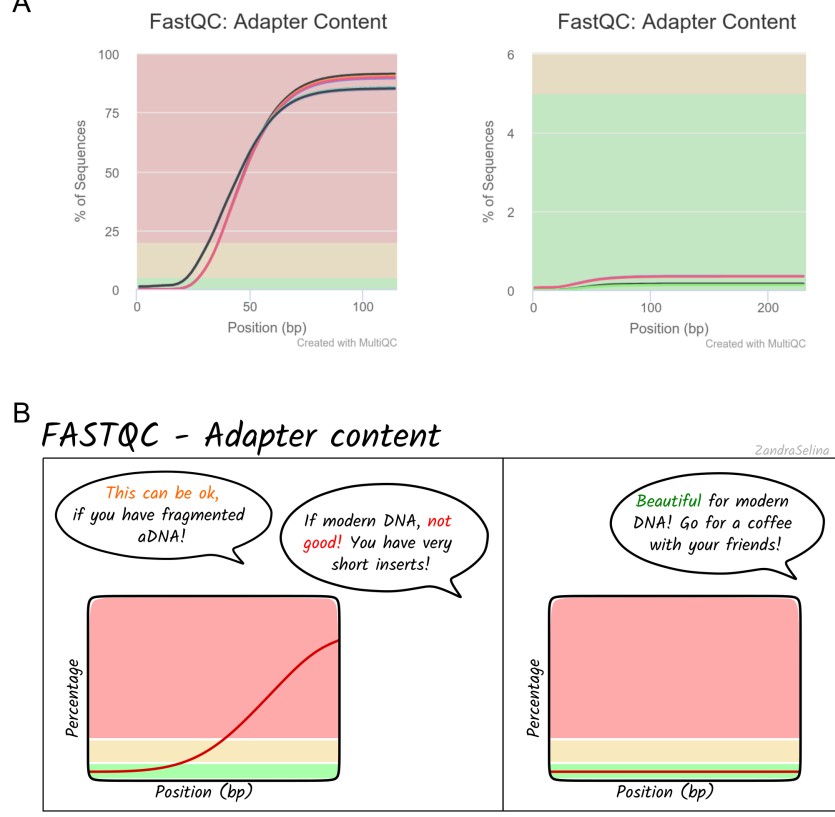

**Figure 5** **Example schematic image of pipeline output documentation with contextual guidance for aDNA.** For each section of the nf-core/eager MultiQC pipeline run report, we provide schematic images that can assist new users in the interpretation of high-throughput sequencing of aDNA. For example, (A) represents MultiQC images of the FastQC report for the amount of sequencing reads with library adapters and (B) is the schematic version counterpart in the nf-core/eager documentation with notes specific for aDNA libraries.

aDNA more accessible to researchers new to the field, by providing practical guidelines on how to evaluate characteristics and effects of aDNA on downstream analyses.

The development of nf-core/eager in Nextflow and the nf-core initiative will also improve open-source development, while ensuring the high quality of community contributions to the pipeline. While Nextflow is written primarily in Groovy, the Nextflow DSL simplifies a number of concepts to an intermediate level that bioinformaticians without Java/Groovy experience can easily access (regardless of own programming language experience). Furthermore, Nextflow places ubiquitous and more widely known command-line interfaces, such as bash, in a prominent position within the code, rather than custom Java code and classes (as in EAGER). We hope this will motivate further bug fixes and feature contributions from the community, to keep the pipeline state-of-the-art and ensure a longer life-cycle. This will also be supported by the open and active nf-core community who provide general guidance and advice on developing Nextflow and nf-core pipelines.

It should be noted that the scope of nf-core/eager is as a generic, initial data processing and screening tool, and not to act as a tool for performing more experimental analyses that requires extensive parameter testing such as modelling. As such, while similar pipelines designed for aDNA have also been released, for example ATLAS (*Link et al., 2017*), these generally have been designed with specific contexts in mind (e.g., human population genetics). We therefore have opted to not include common downstream analysis such as Principal Component Analysis for population genetics, or phylogenetic analysis for microbial genomics, but rather focus on ensuring nf-core/eager produces useful files that can be easily used as input for common but more experimental and specialised downstream analysis. Secondly, given nf-core/eager's broad scope of allowing analysis of different target organisms, default parameters of the pipeline are selected as general 'sensible' defaults to account for typical ancient DNA characteristics - and are not necessarily optimised for every use-case. However, the extensive documentation should help researchers decide which parameter values are most suitable for their research.

## CONCLUSION

nf-core/eager is an efficient, portable, and accessible pipeline for processing and screening ancient (meta)genomic data. This re-implementation of EAGER into Nextflow and nf-core will improve reproducibility and scalability of rapidly increasing aDNA datasets, for both large and small laboratories. Extensive documentation also enables newcomers to the field to get a practical understanding on how to interpret aDNA in the context of NGS data processing. Ultimately, nf-core/eager provides easier access to the latest tools and routine screening analyses commonly used in the field, and sets up the pipeline for remaining at the forefront of palaeogenetic analysis.

## ACKNOWLEDGEMENTS

We thank the nf-core community for general support and suggestions during the writing of the pipeline. We also thank Arielle Munters, Hester van Schalkwyk, Irina Velsko, Katherine Eaton, Luc Venturini, Marcel Keller, Pierre Lindenbaum, Pontus Skoglund, Raphael Eisenhofer, Torsten Günter, Kevin Lord, and Åshild Vågene for bug reports and feature suggestions. We are grateful to the members of the Department of Archaeogenetics at the Max Planck Institute for the Science of Human History who performed beta testing of the pipeline. We thank the aDNA Twitter community for responding to polls regarding design decisions during development. The Gesellschaft für wissenschaftliche Datenverarbeitung (GWDG, Göttingen) kindly provided computational infrastructure for benchmarking. We also want to thank Selina Carlhoff, Maria Spyrou, Elizabeth Nelson, Alexander Herbig and Wolfgang Haak for providing comments and suggestions on this manuscript.

### Funding

James A. Fellows Yates, Thiseas C. Lamnidis., Maxime Borry, Aida Andrades Valtueña, Zandra Fagernäs, and Stephen Clayton were supported by the Max Planck Society. James A. Fellows Yates was supported by the ERC Starting Grant project FoodTransforms (ERC-2015-StG 678901-Food-Transforms) funded by the European Research Council awarded to Philipp W. Stockhammer (Ludwig Maximilian University, Munich). Thiseas C. Lamnidis was supported by the European Union's Horizon 2020 research and innovation programme (grant agreement No 851511) with the ERC Starting Grant project MICROSCOPE funded by the European Research Council awarded to Stephan Schiffels (Max Planck Institute for the Science of Human History). Zandra Fagernäs was supported by the Werner Siemens Stiftung funded project 'Paleobiotechnology' awarded to Christina Warinner (Max Planck Institute for the Science of Human History). Maxime U. Garcia is supported by the BarncancerFonden. There was no additional external funding received for this study. The funders had no role in study design, data collection and analysis, decision to publish, or preparation of the manuscript.

### Grant Disclosures

The following grant information was disclosed by the authors:
Max Planck Society.
ERC Starting Grant project FoodTransforms: ERC-2015-StG 678901-Food-Transforms.
Werner Siemens Stiftung project Paleobiotechnology.
Ludwig Maximilian University, Munich.
European Union's Horizon 2020 Research and Innovation Programme: 851511.
Max Planck Institute for the Science of Human History.
BarncancerFonden.

### Competing Interests

The authors declare there are no competing interests.

### Author Contributions

- James A. Fellows Yates conceived and designed the experiments, performed the experiments, analyzed the data, prepared figures and/or tables, authored or reviewed drafts of the paper, and approved the final draft.
- Thiseas C. Lamnidis, Maxime Borry, Aida Andrades Valtueña, Maxime U. Garcia and Judith Neukamm performed the experiments, authored or reviewed drafts of the paper, and approved the final draft.
- Zandra Fagernäs performed the experiments, prepared figures and/or tables, authored or reviewed drafts of the paper, and approved the final draft.
- Stephen Clayton and Alexander Peltzer conceived and designed the experiments, performed the experiments, authored or reviewed drafts of the paper, and approved the final draft.

## Data Availability

All code is available at GitHub: https://github.com/nf-core/eager.

Code is also archived in Zenodo:

James A. Fellows Yates, Alexander Peltzer, Thiseas C. Lamnidis, Maxime Borry, ZandraFagernas, Aida Andrades Valtueña, …Alex Hübner. (2021, January 14). nf-core/eager: [2.3.1] - Aalen (Patch) - 2021-01-14 (Version 2.3.1). Zenodo. http://doi.org/10.5281/zenodo.4438904.

## Supplemental Information

Supplemental information for this article can be found online at http://dx.doi.org/10.7717/peerj.10947#supplemental-information.

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
