# Peer review of "Reproducible, portable, and efficient ancient genome reconstruction with nf-core/eager"

_PeerJ, doi:10.7717/peerj.10947_

## Round 0.1 · original submission · Minor Revisions

Congratulations on a very good submission highly appreciated by the reviewers. Please prepare a revision reflecting the reviewers comments (including additional explanations about methodological choices). I think the figure suggested by Felix Key is a good idea; I find it hard to judge how much additional effort the inclusion of samtools mpileup would require. If there is a comparison between samtools mpileup and the GATK genotyper in the literature this could be sufficient to put the genotyper performance in relation.

·

Basic reporting

The writing was very good, clear and well explained, it was a really pleasant reading. The references were well chosen and representative of the aDNA field, the introduction and background appropriate and the authors clearly show how nf-core/eager fits and contributes to the development of the aDNA field. The manuscript provides an extensive description of all features included in the software and their purpose and the figures and tables are relevant and good supports of the written content in the article (two minor comments below). The manuscript format is appropriate.

The authors provide extensive supplementary data that allow to fully reproduce all presented results and provide all accession numbers for downloading the public data used in the manuscript.

Regarding the figures, my only comments are the following:
Figure 2: The general idea of the figure is clear, but there are some parts that I don’t understand. For instance, I don’t follow the logic of the library names on the left, and how they are pooled across the pipeline. Maybe these names could be more didactic to better understand why there are different merging points. (Writing question about the title: is it Schematic or Scheme?)

Figure 4: I understand that this image is an example of the many schemes that are generated through the pipelines, but maybe it would be nice to show on the side (e.g., Fig 4b), the real plot that would be obtained for this step, so the reader can understand what the schematic image is explaining.

Experimental design

The experimental design and benchmarking strategies chosen were appropriate to demonstrate the performance of the software, and followed a high technical standard. The motivations for developing this software and the gaps that is looking to fill are also well explained. The pipeline provides an efficient, standardized, reproducible way to run multiple software that are commonly used for processing aDNA data, and it incorporates some new features and optimizations that are currently missing in other similar options currently available (e.g., PALEOMIX or EAGER).

Methods are clearly described with all necessary information to replicate (and on top of this, the GitHub site has extensive extra documentation for every single step that can be run in this pipeline). The authors have really done a remarkable effort in making every aspect of the method clear, transparent and well documented, to facilitate a rapid adoption in the field.

My only minor comment on this is the following:
Lines 234-235: Wouldn’t it make sense for the Benchmarking step to also run it with all default parameters normally used in nf-core/eager. This would allow to see whether results are reproduced regardless of parameter optimization. It would be interested to know if same conclusions are reached even without any a-priori information.

I also have the following suggestion for future releases:
It would be great to incorporate an additional step for metagenomic analyses, to eliminate low-complexity sequences (e.g., with BBMask from the JGI BBTools). This will greatly reduce false positives in downstream Kraken2/MALT analyses.

Validity of the findings

The pipeline improves currently available alternatives, and provides a very useful resource for the ancient DNA field, that can continue growing in the future with new features. The authors provide a thorough explanation of every point that is advantageous in this pipeline, while acknowledging aspects that may be considered disadvantageous for some users, but well justifying why it was done this way. It is definitely a great asset for the field.

My only very minor comment on this is the following:
Regarding the exclusion of mapDamage2 from the aDNA evaluation and modification step, I believe that the rationale behind could be a bit more explained given than many researchers in the field continue to use it (e.g., “more performant successor method is now available”, more performant in terms of what? Is DamageProfiler better at every level? A more detailed explanation would help here). For instance, table 1 indicates that Damage Rescaling is partial in nf-core/eager, so how is this managed?

Additional comments

I liked the article and the pipeline a lot, and I believe that it will be a very useful resource for the ancient DNA field. Most importantly, it is built in such a way that will greatly facilitate a continuous growth in the future, by incorporating new features or even be taken by different labs as starting points for building customized solutions optimized for the lab needs.

One minor comment:
- Line 122 and 145: There are two different references for MultiVCFAnalyzer. Is this correct?

·

Basic reporting

The manuscript is well written in professional English. Nothing further to add.

Experimental design

No analyses of primary data is presented in the study.

Validity of the findings

The paper presents a pipeline utilizing known bioinformatic software to perform basic ancient DNA data processing. The pipeline provides many major advantages to the field in terms of usability, scalability, and reproducibility.

Additional comments

Fellow Yates et al present in their manuscript a one-stop-shop analysis pipeline for initial raw ancient DNA (aDNA) sequencing data processing utilizing a wide range of established bioinformatic tools. It is the successor of eager1 (Peltzer et al 2016) and based on nextflow (an omics-dedicated pipeline language). The improvements include (non-exhaustive list) a proper parallisation, ability to merge different types of data per sample (SE/PE, UDG-treatment), command-line usage (but GUI still available) and a significant extension of the software repertoire. Here, most important probably the addition to perform metagenomic analysis. The software and its configuration files are easily portable, which is an important step towards improved reproducibility and can streamline community access to the backbone results of published manuscripts/data. It targets a user base of both versed or inexperienced aDNA scientists, with the latter often without much/any knowledge in programming. To make up for that gap, they provide extensive documentation incl. info-graphics that simplify the interpretation of graphs. That is an important contribution in itself, given that in particular newcomers to the field who use nf-core/eager are prone to fly blindly through the myriads of (default) parameters that have a tremendous impact on their data and its interpretation/troubleshooting.

I performed a simple test run using a SRA file containing a known ancient organism. Set up of the pipeline was easy on a naive HPC system (minor trouble shooting due to SE data). The different report html’s carry plenty of information including explanations for less experienced users. I might have missed it, but there was no table with alignment-based summary stats (%coverage, mean coverage, % 3’/5’ deamination,…), which I would consider a worthwhile addition.

Overall, while the aDNA field is still a niche (expanding though), I have no doubt that nf-core/eager will take its place as a widely used software for the time being.

I have only a few comments that I recommend to address:
- The authors decided to include the unsupported GATK genotyper. Including an officially abandoned piece of software (especially one that has known biases) seems ill-advised and I suggest to remove it. A simple replacement could be samtools mpileup, which is scalable and together with bcftools generates VCF files for downstream filtering/analyses.
- I recommend to add a main text figure that provides an overview about all the supported bioinformatic tools and how they relate to one another (input/output) and towards what type of analyses/result they lead.


Disclaimer: This review is written by Felix Key. I have co-authored manuscripts with several of the authors stemming from my time at the MPI SHH (’16 - ’18). However, I have no asset of any kind in the nf-core/eager pipeline.

·

Basic reporting

This is a well written paper, with clear logical progression, and informative figures. The introduction provides adequate background about the unique characteristics of aDNA and cites relevant prior literature. The authors do a convincing job of identifying the need for an updated aDNA pipeline, and clearly outline the areas in which nf-core/eager improves and extends upon prior methods.

Experimental design

The authors have done an excellent job of incorporating current aDNA best-practices into the design and workflow of their computational pipeline. Whilst reasonable opinion might differ about specific choices of tooling and the configuration of some flags, nf-core/eager provides most users with an excellent foundation for processing their aDNA data.

Validity of the findings

The authors commitment to clear user documentation, computational reproducibility and implementation of modern software development practices are to be commended. As the authors correctly highlight, the literature is full of “insufficient 'prose' based reporting” of computational methods. Adoption of high-quality standardised pipelines like nf-core/eager are an important step towards improving the validity of finding across the broader aDNA literature.

Additional comments

Line 321: The link to the user documentation (https://nf-co.re/eager/docs) is incorrect
Line 374: Please provide an ftp link to the NCBI screening database
Line 380: The directory containing the demonstration and benchmarking environments is “content/supplement/”

---

## Round 0.2 · accepted · Accept

Thanks for the great response to the reviewers' request. I am happy to accept the manuscript for publication.